# Smartphone-Based Quantitative Detection of Ochratoxin A in Wheat via a Lateral Flow Assay

**DOI:** 10.3390/foods12030431

**Published:** 2023-01-17

**Authors:** Yunxin Tian, Xiaofeng Hu, Jun Jiang, Xiaoqian Tang, Zhiquan Tian, Zhaowei Zhang, Peiwu Li

**Affiliations:** 1College of Ecology and Environmental Sciences, Tibet University, Lhasa 850000, China; 2Hubei Hongshan Laboratory, Wuhan 430070, China; 3National Reference Laboratory for Agricultural Testing (Biotoxin), Key Laboratory of Detection for Mycotoxins, Ministry of Agriculture and Rural Affairs, Oil Crops Research Institute of Chinese Academy of Agricultural Sciences, Wuhan 430062, China; 4College of Chemistry and Molecular Sciences, Wuhan University, Wuhan 430072, China

**Keywords:** Ochratoxin A, smartphone, Europium nanospheres, fluorescent lateral flow test strip, wheat, quantitative detection

## Abstract

Ochratoxin A (OTA) poses a severe health risk to livestock along the food chain. Moreover, according to the International Agency for Research on Cancer, it is also categorized as being possibly carcinogenic to humans. The lack of intelligent point-of-care test (POCT) methods restricts its early detection and prevention. This work establishes a smartphone-enabled point-of-care test for OTA detection via a fluorescent lateral flow assay within 6 min. By using a smartphone and portable reader, the assay allows for the recording and sharing of the detection results in a cloud database. This intelligent POCT provided (iPOCT) a linearity range of 0.1–3.0 ng/mL and a limit of detection (LOD) of 0.02 ng/mL (0.32 µg/kg in wheat). By spiking OTA in blank wheat samples, the recoveries were 89.1–120.4%, with a relative standard deviation (RSD) between 3.9–9.1%. The repeatability and reproducibility were 94.2–101.7% and 94.6–103.4%, respectively. This work provides a promising intelligent POCT method for food safety.

## 1. Introduction

As an essential crop, wheat is the staple food for 35% of the world’s population, and China currently leads the world in wheat cultivation [1]. OTA, which is present in wheat and metabolized by *Aspergillus ochraceus*, is a major risk factor throughout the food chain. OTA is severely neurotoxic, nephrotoxic, immunosuppressive, and teratogenic in animals and humans. It has been reported that OTA was detected at higher levels in animal and human blood in outbreak areas than in non-outbreak areas [2]. OTA threatens liver and kidney health in animals [3] and eventually harms humans after digesting OTA-contaminated food [4].

OTA has been detected in food and agricultural products in many countries around the world, including China, the USA, Ireland, Portugal, and Pakistan [5]. The test results of roasted and instant coffee collected in the Czech Republic between 2016–2018 showed that the OTA contamination rate reached 80.6% [6]. Therefore, OTA has a strong ability to contaminate and cause economic loss [7]. OTA is physically stable and thus difficult to prevent and control. Detecting OTA as early as possible can help to prevent OTA from contaminating food and feed.

In addition to the official detection methods, such as gas chromatography-tandem mass spectrometry (GC–MS/MS) [8], high-performance liquid chromatography (HPLC) [9] and liquid chromatography-tandem mass spectrometry (LC–MS/MS) [10], the lateral flow assay, is a growing rapid detection method in OTA detection [11]. Fadlalla et al. [12] used colloidal gold, gold nanoflower strips, and ELISA to detect OTA and OTB. ELISA was highly sensitive to OTA and OTB with an LOD of 0.06 ng/mL. However, ELISA has many steps, and the sensitivity of colloidal gold is low. The LODs of colloidal gold nanoparticles and nanoflower gold strips were 5 μg/mL and 1 μg/mL, respectively [12]. Wu et al. [13] used an immunoassay test strip based on upconversion materials to detect OTA. Although it has pioneered a new path for fluorescence immunochromatography, the incubation time is long [13]. Cheng et al. [14] used ELISA, LC–MS/MS, and a fluorescent microsphere immunochromatographic test strip (FM-ICTS) to detect OTA. Similar results were obtained for all three methods [14]. Sensitivity and stability are two aspects of OTA detection. Colloidal gold, upconversion nanoparticles, and quantum dots (QDs) are typical labels. However, colloidal gold is less sensitive [15], upconversion materials are toxic and expensive [16], and some QDs are toxic and unstable [17]. To address these issues, we used Eu-nanospheres as a labelling agent to meet the requirements of sensitivity and stability. Additionally, current methods rely on desk instruments with unsatisfactory real-time data processing and difficult transportation. A smartphone-assisted detection method can satisfy point-of-care testing (POCT) through its camera function and data processing. POCT has the advantages of rapidness, simplicity, and reliability. A smartphone can directly serve as a camera, thus simplifying the analytical step [18]. At the same time, it is difficult to apply a smartphone camera due to different camera functions and cellphone models [19].

In this study, we designed a fluorescent POCT based on Eu-nanospheres. It was a competitive assay which consists of a portable reader and lateral flow test strip. The portable reader, with a separated camera, was produced by 3D printing technology and can be connected to a smartphone for the rapid analysis of the strips. The app can collect RGB values to convert into grayscale values to represent fluorescent intensity (I). Europium (Eu) is an element with high quantum yield, narrow bandwidth, long-lived emission, large Stokes shifts, low toxicity, etc., which is widely used in immunoassay [20]. The Eu element can effectively avoid interference from background fluorescence. The fluorescent oxide latex nanospheres prepared with the Eu element (Eu-nanospheres) has many advantages, included bright colors, uniform particle size, good monodispersity, high reproducibility of test results, and convenient use. The Eu-nanospheres were coupled to a monoclonal antibody as a marker, and OTA and sheep anti-mouse IgG, coupled with bovine serum protein, were fixed to the strip as the control line (C line) and test line (T line), respectively. Moreover, a homemade rapid quantification reader and a cell phone were used. This work provides a promising POCT method for food safety by using alternating antibodies.

## 2. Materials and Methods

### 2.1. Materials and Reagents

Nanospheres of monodisperse polystyrene modified by carboxyl groups were acquired from Shanghai Uni Biotechnology Company Limited (SHH, CHN). We used a Millipore Milli-Q system to obtain ultrapure water. The goat anti-mouse immunoglobulin (IgG) was acquired from the Wuhan Boster Biological Technology Company Limited (WUH, HB, CHN). Europium oxide latex nanospheres, deoxynivalenol (DON), standard AFB1, diacetoxyscirpenol (DAS), ochratoxin A (OTA), fumonisin B1 (FB1), heat-stable enterotoxin (ST), T-2 toxin, zearalenone (ZEN), bovine serum albumin (BSA), disodium phosphate, boric acid, sodium chloride, Ochratoxin -A-bovine serum albumun (OTA-BSA), borax, potassium dihydrogen phosphate, methanol (MeOH), sucrose, polyvinylpyrrolidone K30 (PVPK30), acetonitrile (ACN), Tween-20 and *N*-(3-dimethylaminopropyl)-*N*′-ethylcarbodiimide hydrochloride (EDC) were obtained from the Sigma–Aldrich Company Limited (St. Louis, MO, USA). The sample pad (GFCP000800 glass fibres, fusion three, and fusion five), absorbent pad, HFC13502 nitrocellulose (NC) membrane, and plastic adhesive card were purchased from Millipore (Billerica, MA, USA). The FF120HP NC membrane was acquired from Whatman International Ltd. (Maidstone, Kent, UK), and the CN 95 NC membrane was acquired from Sartorius (Goettingen, NDS, Germany).

### 2.2. Apparatus

We purchased a CF16RX high-speed refrigerated centrifuge from HIT (Tokyo, Japan). The vortex oscillator was acquired from VELP Corp. (Bohemia, NY, USA). The detection strips were prepared using a CM 4000 guillotine cutter, XYZ 3050 dispensing platform, and batch laminator (LM 4000), which were acquired from BioDot Inc. (Irvine, CA, USA).

### 2.3. Development of the POCT Portable Reader

Benefiting from the development of smartphone technology, smartphones can also be used for fluorescent immunoassay strip data analysis after installing the appropriate application [21]. Therefore, we created a portable reader that can be used with smartphones (Figure 1). A commercial power supply is used to power the reader, and a 3-in-1 connector is used, which allows for compatibility with smartphones with three types of data cables (type-c, lightning, and micro-USB). First, the smartphone is connected; then, the app is opened. Next, the card reader is powered on, the test strip is inserted into the slot, and the card slot is simply inserted into the main body. Once inserted into the slot, the LED light source (365 nm) excites the Eu-nanospheres to emit fluorescence (613 nm); the fluorescence signal passes through the filter and is accepted by the camera and transmitted to the phone (Figure 1). Finally, the RGB values were converted to grayscale values for quantitation according to (I) = 0.299R + 0.587G + 0.114B [22].

### 2.4. Preparation of the Anti-OTA Monoclonal Antibody

We developed a monoclonal antibody against OTA, according to our previous report [23]. The OTA-BSA was injected into several subcutaneous sites in BALB/c mice (7–8 weeks old) to prepare an anti-OTA monoclonal antibody. Then, the OTA-positive cells were screened again to ensure their monoclonal stability, and the stable monoclonal antibody was amplified. Freund’s incomplete adjuvant (0.3 mL) and the resulting hybridoma cells by amplification were injected into the BALB/c mice. After that, we used caprylic acid-ammonium sulfate to purify the extracted ascites residue via an immunoaffinity column (IAC) with protein G, which was subsequently stored in the frozen state (−20 °C).

### 2.5. Conjugation of OTA mAb and Eu-Nanospheres (EuNPs)

Europium oxide latex nanospheres for OTA monoclonal antibodies are white emulsions under visible light and bright orange under UV light, which have an excitation wavelength of 365 nm, an emission wavelength of 613 nm, and a particle size of approximately 200 nm. The OTA monoclonal antibody was coupled to europium oxide latex nanospheres as follows. First, 200 µL of europium oxide emulsion was added to 800 µL of 0.2 mol/L pH 8.0 borate buffer and shaken. Then, the mixture was well mixed via an ultrasonic cell disruptor for 4 s. Then, a volume of 30 mg/mL EDC solution (20 µL, 40 µL, 60 µL, and 80 µL) was added; the mixture was vortexed and centrifuged; and the supernatant was discarded. After resuspending the precipitate in borate buffer, the precipitate was shaken and mixed and then sonicated with an ultrasonic cell disruptor. Next, 20 µL of OTA monoclonal antibody solution (10 µL, 20 µL, 40 µL, and 80 µL) was added, and the mixture was vortexed, mixed, rotated and mixed for 12 h at 10 °C. The mixture was centrifuged for 10 min (13,300 r/min, 10 °C), and the supernatant was discarded. Finally, 1 mL of 0.5% BSA solution was added to resuspend the sample (the remaining active sites of the europium oxide latex nanospheres were closed by the protein), which was vortexed, mixed, rotated and mixed for 2 h at 10 °C (250 r/min) and stockpiled at 4 °C in the fridge until use.

### 2.6. Characterization of the mAb@Eu-Nanospheres

The anti-OTA mAb conjugated with the Eu-Nanospheres was proven via zeta potential distribution and size distribution by intensity. Briefly, 50 µL of Eu-Nanospheres and mAb@Eu-Nanospheres were diluted to 1 mL separately for the characterization of the mAb@Eu-Nanospheres.

### 2.7. iPOCT Assembly

The components of the iPOCT contained 4 parts: the plastic backing pad, absorbent pad, NC membrane, and sample pad. One side of the plastic pad was attached by the whatman FF120 NC membrane (Sartorius CN95, whatman FF120, and Millipore HF135), while the fusion 5 sample pad (fusion 3, fusion 5 and blood filter membrane) and absorbent pad were overlaid on it, with the adjacent pads overlapping by 1 mm at the junction. The test part was based on the NC membrane, with horizontal C and T lines from the bottom to the top. The T line on the NC membrane was wrapped at dispensing rates of 0.3, 0.5 and 0.7 µL/cm with 0.5 mg/mL of OTA-BSA (0.2, 0.3, 0.4, 0.5, 0.6 and 0.7 mg/mL), and then the C line on the NC membrane was wrapped at dispensing rates of 0.3, 0.5 and 0.7 µL/cm with 0.2 mg/mL of goat anti-mouse IgG (0.05, 0.1, 0.2, 0.3, 0.4 and 0.5 mg/mL). After spraying, the iPOCT were dried in an oven at 37 °C for 1 h, removed, cut into strips of 4 mm width and finally stockpiled in a sealed fridge at 4 °C.

### 2.8. Preparation of the Running Buffer

Running buffer was used to ensure the sample was distributed on the NC membrane uniformly. First, 3 mL of 0.5% BSA + 1% sucrose + 0.5% polyvinylpyrrolidone + 2.5% Tween-20 aqueous solution of running buffer (2.5% Tween-20 aqueous solution, 1% sucrose + 2.5% Tween-20 aqueous solution, 1% sucrose + 2.5% Tween-20 + 0.5% BSA aqueous solution) was added to treat the sample. Then, 600 µL running buffer was added into 100 µL sample before the test.

### 2.9. Sample Treatment

Twenty-five grams of ground wheat powder was homogenized with 100 mL acetonitrile/water (60:40, *v*:*v*) at 15,000 r/min for 3 min. After filtering, 1 mL of filtrate was mixed with 3 mL running buffer (containing 0.5% BSA, 1% sucrose, 0.5% polyvinylpyrrolidone, and 2.5% Tween-20) before use.

### 2.10. Validation of the POCT

The LOD, calibration curve, linear range, repeatability, reproducibility, stability, and specificity were verified using spiked wheat samples. For the calibration curve, we added multiple concentrations of standards (0.1, 0.15, 0.2, 0.5, 0.75, 1, 1.2, 1.5, 2, and 3 ng/mL) to the wheat samples for detection; then, after analyzing the iPOCT using a portable fluorescence reader, we constructed the relationship Y = ax + b, where the logarithm of the OTA concentration is represented by X and the intensity specific value of the T and C line fluorescence is represented by Y. Regarding the LOD, 21 blank wheat samples were tested, and it was calculated via the formula “LOD = 3σ/s”, where σ is the standard deviation of the 21 blanks and s is the slope of the calibration curve [24]. For the coefficient of variation (CV), six iPOCTs from different batches and six iPOCTs from the same batch were used to verify the interassay and intraassay variability. The final coefficient of variation was analyzed to determine the variability. To verify the specificity, OTA, ZEN, AFB1, T-2, FB1, DON, ST, and DAS were tested, and the specificity of the strip for OTA was verified by comparing the T/C obtained for each test. The concentration of each mycotoxin was 1 ng/mL, and three parallel sets of experiments were performed for each species.

The iPOCT validation was conducted using three wheat samples spiked by OTA at levels of 0.1, 0.5, 1, 1.5, and 2 ng/mL, respectively. The same samples were determined by using UPLC-MS/MS method. A Thermo C18 column (100 mm × 2.1 mm, 3 μM) at 30 °C was used for chromatographic separation. Mobile phase A was water (0.1% formic acid and 5 mmol/L ammonium formate), and mobile phase B was acetonitrile aqueous solution (0.1% formic acid and 5 mmol/L ammonium formate). The elution flow rate was 0.2 mL/min, and the injection volume was 1 μL. The gradient elution program were phase A and phase B: 0–2 min, 65–5% A, 35–95% B; 2–5 min, 5% A, 95% B; 5–5.1 min, 5–65% A, 95–35% B; 5.1–6 min, 65% A, 35% B. The MS/MS parameters are shown in Table A1.

## 3. Results and Discussion

### 3.1. Optimization of mAb@Eu-Nanospheres

During labelling, in the Eu-nanospheres with OTA monoclonal antibodies, greater EDC consumption was supposed to activate more Eu-nanospheres. Typically, the same or higher molar amount of EDC was requested than the molar amount of the carboxyl groups on the Eu nanospheres. The results found that 40 µL EDC was the optimized dose to ensure the full activation of the Eu-nanospheres. For optimizing the anti-OTA mAb concentration, decreasing the anti-OTA mAb concentration enhanced the detection limit but caused a lower fluorescence intensity. We observed that 20 µL of 1 mg/mL anti-OTA mAb was preferred to ensure clear T and C lines with little background interference.

We found that the zeta potential distribution was decreased from −62.4 mv to −30.5 mv, and the size distribution increased from 100.7 r.nm to 119.4 r.nm after conjugating the anti-OTA mAb and Eu nanospheres (Figure A1).

### 3.2. Optimization of iPOCT

The parameters were optimized for better fluorescence intensity of the T and C lines of the iPOCT, including the sample pad, NC membrane, IgG concentration, and OTA antigen concentration. We evaluated the testing performance via the fluorescence signal on the T and C lines. Then, we chose: (a) a fusion5 sample pad; (b) a Whatman FF120 NC membrane, 0.2 mg/mL IgG, 0.5 mg/mL OTA-BSA, to ensure the fluorescence intensity of the T and C lines. A proper running buffer promised an even more reliable lateral flow and acceptable iPOCT results. After comparing four running buffers, the one containing 0.5% BSA, 1% sucrose, 0.5% polyvinylpyrrolidone, and 2.5% Tween-20 was used for further experiments.

### 3.3. Procedure of the iPOCT

Before the test, the iPOCT was removed and brought to room temperature [25]. First, 600 µL of running buffer and 100 µL samples were added to the microplate and mixed well. Then, the strips were mounted in the microplate for 6 min of incubation (Figure 2a). If the sample is OTA positive, the free mAb@Eu-nanospheres will react in the microplate to form an immune complex that is not caught by the OTA-BSA sprayed at the T line, but is caught by the goat anti-mouse immunoglobulin IgG at the C line. Thus, under 365 nm UV excitation, the T line’s fluorescence intensity diminishes or disappears, while the C line exhibits violet–red fluorescence. If the sample is OTA negative, the T line captures some of the free mAb@Eu-nanospheres. The C line captures the remaining free MAb@Eu-nanospheres, and both the T and C lines exhibit violet fluorescence under UV excitation. In the end, the test strip was tested by inserting it into the POCT strip portable reader (Figure 1 and Figure 2b). Twenty pixel points were evenly selected in each of the areas of the T and C lines in the fluorescent image; then, their RGB values were collected via the app and converted to greyscale values using the formula. The average of the greyscale values represents the fluorescence intensity of the T and C lines.

### 3.4. Validation of the POCT

The LOD, linear range, repeatability, and reproducibility were detected through spiking experiments. Five repeated experiments were performed on 21 blank wheat samples (Table A2) to determine the LOD. The mean value was recorded to calculate the standard deviation. The slope of the standard curve could be calculated based on the standard curve in the matrix. Finally, the above data could be combined to calculate the LOD in a wheat matrix as 0.02 ng/mL, and the linear range of OTA was 0.1~3 ng/mL (Figure 3a). Compared with the previous methods via lateral flow test strip with [26] or without a smartphone [13,14,27], our LOD was up to 5 times higher with the lesser assay time of 4 min. In order to prove that our method could be used for the rapid and sensitive detection of OTA, the specificity of the POCT strip was verified by comparing the fluorescence signals of eight different mycotoxins: OTA, AFB1, ZEN, T-2, DON, FB1, ST, and DAS (Figure 3e). Each mycotoxin was prepared at 1 ng/mL for specificity validation, and the comparison showed that the POCT strip had great selectivity for OTA. The reproducibility was verified by performing an interassay recovery procedure on six batches of the POCT strips (Figure 3b). The OTA recoveries were 94.6–103.4%, and the CVs were 3.5%. The reproducibility was then verified by using the same batch of POCT strips (Figure 3c) and performing six assays on the same spiked wheat samples. The OTA recoveries were found to be 94.2–101.7%, with CVs of 2.7%. To verify the stability, the POCT strip was stockpiled in a fridge at 4 °C and removed every 30 days for testing (Figure 3d). The test recoveries ranged between 95.5% and 102.3%, which shows that the POCT strip method has good stability over 180 days. The specificity of the POCT strip was verified by spiking eight different mycotoxins: OTA, AFB1, ZEN, T-2, DON, FB1, ST, and DAS (Figure 3e). Each mycotoxin was prepared at 1 ng/mL for specificity validation, and the comparison showed that the POCT strip had great selectivity for OTA. The accuracy in the real samples was further verified by spiked recovery experiments on three wheat samples via iPOCT and UPLC-MS/MS (Table 1). The results of the UPLC-MS/MS and iPOCT suggested that the iPOCT can be an alternative to the UPLC-MS/MS. Under optimal conditions, the POCT strip recoveries ranged between 89.1% and 120.4%, with an RSD of between 3.6% and 9.1%.

Most of the reported readers work directly with the smartphone camera as the imaging element, but different smartphone cameras lead to different imaging results, which may lead to deviation. This is why the reader is designed with a separate camera: an HD camera inside the reader and a 3-in-1 cable to connect to the smartphone. The photodiode is connected to the smartphone by the 3-in-1 cable. Therefore, the smartphone could detect the fluorescence as soon as the Mab@Eu-nanospheres are irradiated by the photodiode via UV light.

## 4. Conclusions

This study reports a sensitive, rapid, and convenient method for detecting OTA in wheat. The method included an iPOCT and a portable reader. The fluorescence iPOCT depended on a competitive immunochromatographic strip. The turnaround time was down to six minutes to ensure testing reliability. We recorded a low LOD of 0.02 ng/mL, wide linearity of 0.1–3 ng/mL, and considerable recovery of 89.1–120.4%, with satisfied repeatability and reproducibility of 94.2–101.7% and 94.6–103.4%. Consequently, this iPOCT can be used for on-site screening and OTA detection. A future trend in this field of study could include developing the complex matrix’s highly sensitive, high throughput POCT method.

## Figures and Tables

**Figure 1 foods-12-00431-f001:**
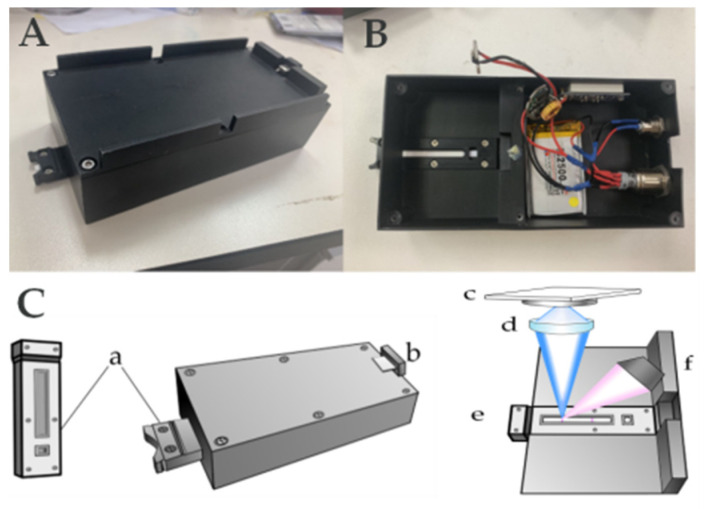
(**A**) The external view of iPOCT reader; (**B**) top view of iPOCT reader; (**C**) schematic illustration of the POCT reader: (a) test strip slot, (b) 3-in-1 data cable (type-c, lightning, and micro-USB), (c) photodiode, (d) filter, (e) UV light (365 nm), and (f) test strip slot.

**Figure 2 foods-12-00431-f002:**
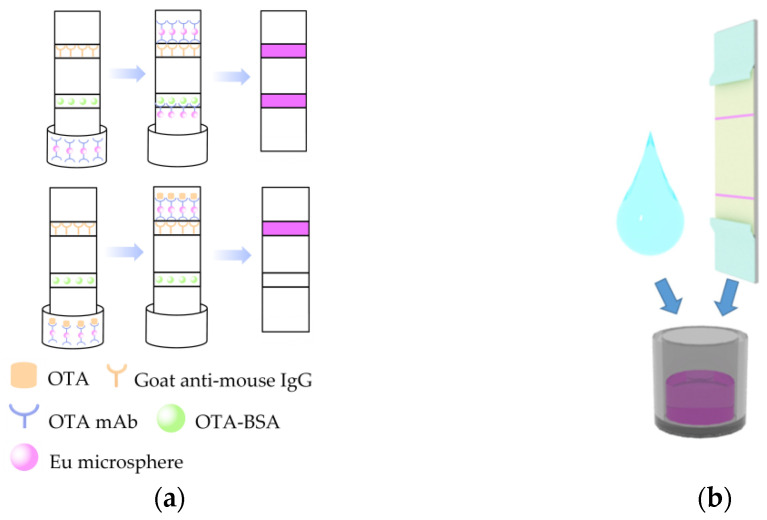
(**a**) Schematic illustration of the iPOCT strips. (**b**) The solid phase MAb@Eu-nanospheres in the well microplate were prepared by freeze-drying the reaction products after adding 20 µL of OTA-Mab (1 mg/mL) into 200 µL of Eu-nanospheres.

**Figure 3 foods-12-00431-f003:**
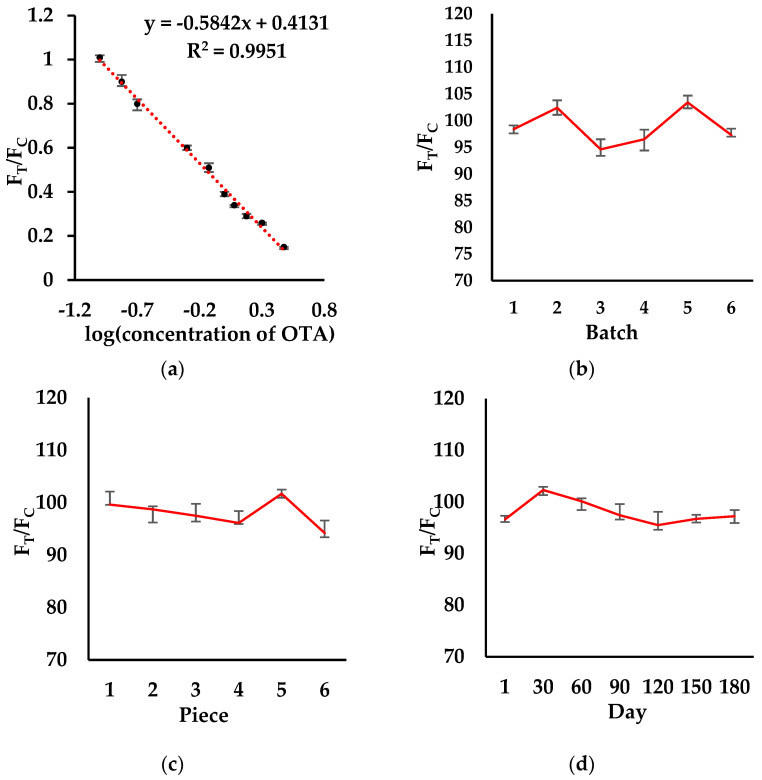
OTA calibration curves (**a**), interassay recovery batch reproducibility (**b**), intraassay recovery batch repeatability (**c**), 180-day stability (**d**), and specificity aflatoxin B1 (AFB1), zearalenone (ZEN), T-2 toxins, deoxynivalenol (DON), fumonisin B1 (FB1), sterigmatocystin (ST), and diacetoxyscirpenol (DAS) (**e**). Each data represented the average value from three measurements in wheat.

**Table 1 foods-12-00431-t001:** Comparison between this proposed iPOCT and UPLC-MS/MS method using wheat.

Sample	OTA Spiked Level (ng/mL)	OTA Found (ng/mL)		RSD
		UPLC-MS/MS	iPOCT	
	0.1	0.096	0.112	9.1
	0.5	0.501	0.487	5.4
Wheat	1	1.022	1.028	6.7
	1.5	1.503	1.507	4.7
	2	2.030	2.057	3.6

## Data Availability

Data is contained within the article.

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
