# Peer review of "Smartphone-Based Quantitative Detection of Ochratoxin A in Wheat via a Lateral Flow Assay"

_foods, 2023, doi:10.3390/foods12030431_

Round 1

Reviewer 1 Report (Previous Reviewer 2)

The manuscript has improved a lot. All questions were answered adequately.

Author Response

Reviewer 2 Report (New Reviewer)

The paper describes the detection and quantitation of OTA using a smart-phone  based method  via a lateral flow assay. Although I find that most of the work within the paper thoroughly covers the various aspects of the study, there is still some work required to improve the structure and description of the work. Particularly method comparisons and  RSD’s etc. I particularly miss tabulated comparisons and a more thorough discussion on previous methods.

1)      It would be good to include a true picture of the device.

2)      Has this sort of device been used for the detection of other toxins?

3)      Please address formatting etc. such as spaces prior to citation. The abstract has extra spaces etc.

4)      Figure 3 caption is poorly written. Maybe something like this: OTA calibration (a) and interassay recovery batch reproducibility (b)….

5)      You have not described ‘Eu’ in the introduction.

6)      Describe the function of the smart phone, the software or app, briefly in the introduction and maybe more detail in the discussion ?

7)      Why are the results for the HPLC/MS in the appendix? For the Table A.2. if there were more replicates, that needs to be described and RSD’s for each test needs to be described.

8)      Line 206: were there technical replicates?

9)      There is no reference to Table A.2.?

10)   Appendix A.5 – do you mean ‘material data sheet’

11)   Line 200:  Use ‘Coefficient of variation (CV)’ in method as it is later abbreviated.

12)   I don’t really like how Table 1 is structured. Its almost as if a table is not required and could be described in text.

Author Response

This manuscript is a resubmission of an earlier submission. The following is a list of the peer review reports and author responses from that submission.

Round 1

Reviewer 1 Report

The work is interesting

my comments below:

1- include a brief description of POCT in the introduction

2- Incorporate results of chemical characterization of nanospheres functionalized with antibodies.

3- line 157: what concentration of OTA-BSA was put on the membrane?

4. Better explain point 2.8

5- in point 2.9 of validation: Was a comparison of the quantification made with a traditional method by HPLC-MS?

Author Response

Dear Editor,

Thank you very much for your comments and suggestions.

Please allow me to response the comments one by one.

1.include a brief description of POCT in the introduction.

Respnse 1: According to your suggestion, I had already added a description of POCT in line 64 in introduction.

2.Incorporate results of chemical characterization of nanospheres functionalized with antibodies.

Response 2: The Zeta Potential Distribution and Size Distribution by Intensity were added into the Appendix in line 328.

3- line 157: what concentration of OTA-BSA was put on the membrane?

Response3: The concentration of OTA-BSA put on the membrane was 0.5mg/mL.

4. Better explain point 2.8

Response 4: I had re-explained the point 2.8 under your suggestion.

5- in point 2.9 of validation: Was a comparison of the quantification made with a traditional method by HPLC-MS?

Response 5: I made a comparison with UPLC-MS/MS and the data was added into the Appendix in line 334.

Special thanks to you for your good comments.

Kind regards,

Yunxin Tian

Reviewer 2 Report

The paper focuses on an important problem that exists all over the world. Mycotoxins, especially OTA are stable and very dangerous compounds that can not be removed from the foods and animal feeds. Therefore, development of a quantitative and sensitive POCT assay coupled to a smart phone as the detection unit is a new and very useful investigation. Basically, the paper is well written and organized but there are some minor points to explain or to correct.

69 IGG is better to write as IgG and also the sentence should be rephrased (not cultured in bovine serum protein)

92 freezing centrifuge is not a correct term

126 althogh there is a reference for raising the monoclonal ABs but the authors did not describe how they combined OTA with albumin (not in the cited reference as well)

150 not freezer but fridge

160 not freezer and not 0.4oC I guess

167-168 the applied method is not precisely described (homogenization with what? what was the diluent for the filtrate?)

173-174 if the OTA series's concentration was known how it was measured? these are called samples not standards

Why the authors did not mention that their method is a competitive assay? 

It is not written if the mebranes were blocked before their usage or not. If not, why?

Author Response

Dear Editor,

Thank you very much for your comments and suggestions.

Please allow me to response the comments one by one.

1.69 IGG is better to write as IgG and also the sentence should be rephrased (not cultured in bovine serum protein)

Response 1: According to your comments, I had changed IGG into IgG and the sentence was rephrased.

2. 92 freezing centrifuge is not a correct term

Response 2: I had changed the freezing centrifuge into refrigerated centrifuge.

3. 126 althogh there is a reference for raising the monoclonal ABs but the authors did not describe how they combined OTA with albumin (not in the cited reference as well)

Response 3: The OTA-BSA was purchased from Sigma-Aldrich.

4. 150 not freezer but fridge

Response 4: I had already changed the freezer into fridge.

5. 160 not freezer and not 0.4oC I guess

Response 5: I'm very sorry for my negligence of this mistake.The strips were stockpiled at 4 °C.

6. 167-168 the applied method is not precisely described (homogenization with what? what was the diluent for the filtrate?)

Response 6: I had re-written this part according to your comments. 

7. 173-174 if the OTA series's concentration was known how it was measured? these are called samples not standards

Response 7: For the calibration curves , multiple concentrations of standards was added to the  wheat samples for detection. The series's concentration in line 173-174 was the standards'.

8. Why the authors did not mention that their method is a competitive assay?

Response 8: Sorry for my carelessness, i had added 'competitive assay' into the introduction (line 68) and conclusion (line 303).

9. It is not written if the mebranes were blocked before their usage or not. If not, why?

Response 9: The mebranes were blocked by the whatman company.

Special thanks to you for your good comments.

Kind regards,

Yunxin Tian
